# Adaptation Resources and Responses to Wildfire Smoke and Other Forms of Air Pollution in Low-Income Urban Settings: A Mixed-Methods Study

**DOI:** 10.3390/ijerph20075393

**Published:** 2023-04-04

**Authors:** Lawrence A. Palinkas, Jessenia De Leon, Kexin Yu, Erika Salinas, Cecilia Fernandez, Jill Johnston, Md Mostafijur Rahman, Sam J. Silva, Michael Hurlburt, Rob S. McConnell, Erika Garcia

**Affiliations:** 1Suzanne Dworak-Peck School of Social Work, University of Southern California, Los Angeles, CA 90089, USA; 2Department of Population and Public Health Sciences, Keck School of Medicine, University of Southern California, Los Angeles, CA 90089, USA; 3Department of Neurology, Oregon Health Sciences University, Portland, OR 97239, USA; 4Department of Earth Sciences, Dornsife College of Letters, Arts and Sciences, University of Southern California, Los Angeles, CA 90089, USA

**Keywords:** air pollution, wildfire smoke, climate change, adaptation behaviors, health effects, health equity

## Abstract

Little is known about how low-income residents of urban communities engage their knowledge, attitudes, behaviors, and resources to mitigate the health impacts of wildfire smoke and other forms of air pollution. We interviewed 40 adults in Los Angeles, California, to explore their threat assessments of days of poor air quality, adaptation resources and behaviors, and the impacts of air pollution and wildfire smoke on physical and mental health. Participants resided in census tracts that were disproportionately burdened by air pollution and socioeconomic vulnerability. All participants reported experiencing days of poor air quality due primarily to wildfire smoke. Sixty percent received advanced warnings of days of poor air quality or routinely monitored air quality via cell phone apps or news broadcasts. Adaptation behaviors included remaining indoors, circulating indoor air, and wearing face masks when outdoors. Most (82.5%) of the participants reported some physical or mental health problem or symptom during days of poor air quality, but several indicated that symptom severity was mitigated by their adaptive behaviors. Although low-income residents perceive themselves to be at risk for the physical and mental health impacts of air pollution, they have also adapted to that risk with limited resources.

## 1. Introduction

Over the past two hundred years, levels of carbon dioxide in the atmosphere have increased by 50 percent [1], leading to an increase in ambient temperatures, frequency and severity of extreme weather events, and long-term changes in the environment [2]. In turn, although not a cause of air pollution, increases in ambient air temperatures have led to greater concentrations of airborne pollutants such as ozone (O_3_) [3,4,5,6] and fine particulate matter (PM_2.5_) [7,8] in many parts of the world, especially in wildfire-prone areas [9]; however, the associations between temperature and levels of O_3_ and PM_2.5_ exhibit regional variability [10]. Warmer temperatures combined with prolonged drought have also contributed to an increase in the frequency and severity of wildfires that release toxic smoke, especially in the Western United States and Canada [11,12,13,14]. Low-income residents living in urban settings are especially susceptible to air pollution [15,16,17] and to wildfire smoke that can travel hundreds of miles from its origin [18,19]. These residents are also vulnerable to indoor pollution linked to wildfires [20,21,22] and other forms of outdoor pollution due to increased poverty and substandard housing [23,24]. Although the indoor environment is often overlooked in relation to environmental health in general [25] and climate-related health in particular, studies have found that between 80 and 90 percent of the time is spent indoors [26], and this is likely to increase during wildfires due to public health warnings about the risk of exposure to smoke [22].

The physical and mental health impacts of ambient indoor and outdoor air pollution in general, and wildfire smoke, in particular, have been extensively documented [27,28,29,30,31,32], with older adults [33] and low-income communities of color [34,35] being especially susceptible. However, relatively little research has been devoted to understanding the adaptive capacity of low-income urban residents to wildfire smoke and other forms of air pollution. A qualitative study of the impact of wildfire smoke on the mental health and well-being of residents in a predominately rural area of Washington State conducted by Humphreys and colleagues [36] identified opportunities for adaptation and provided recommendations for improving adaptation but did not examine current adaptation resources and responses. Such information is critical to determining the needs of vulnerable populations in urban settings and addressing them through policy and practice, especially those living in urban heat islands where the combination of warm temperatures and air pollution increases the risk of mortality [37], especially in older adults [38].

To address this lack of information, we conducted a qualitative study of adaptation resources and behaviors of households in inner-city neighborhoods in Los Angeles, California, designated as socioeconomic and environmentally “vulnerability communities”. An earlier study of these communities explored residents’ resources and behaviors for adapting to heat waves, the impacts of heat waves on physical and mental health, and threat assessments of future heat waves [39]. In this follow-up investigation, our aims were to explore the following: (1) the extent to which residents of vulnerable urban communities perceive wildfire smoke and other forms of air pollution to be a threat to their health and well-being; (2) the exercise of adaptive behaviors to mitigate air pollution exposure and impacts; (3) availability of resources to enable such adaptation; and (4) the impact of such events and the adaptation behaviors and resources on the physical and mental health of these residents.

## 2. Materials and Methods

### 2.1. Design

In this study, a simultaneous convergent/sampling (quan + QUAL) mixed method design [40] was used to assess socioeconomic and environmental vulnerability to air pollution. We operated from a conceptual framework [31] that places multi-step causal chains associated with climate change within a context of socioeconomic and demographic factors, societal actions, and other non-climate drivers. Drawing from the Integrated Climate Change and Health Indicator Systems Framework [41], the 4R framework has four components: risk of exposure to hazardous environmental conditions, in this case, days of poor air quality; responses or adaptive behaviors; resources to support adaptive capacity; and results of the health impacts of days of poor air quality. Both responses and resources may be framed within a socioecological model that includes individual, family, organization, and community responses and individual physiological, home, and immediate social environment, neighborhood, and mesoscale climate resources.

### 2.2. Participants

Participants for the project were 40 parents of middle and high school students participating in the University of Southern California’s (USC) McMorrow Neighborhood Academic Initiative (NAI), a college prep program that prepares students from South and East Los Angeles for admission to a college or university. Eligibility for participation in the study included living or working in one of the neighborhoods with schools attended by NAI students and having lived in the neighborhood for a minimum of 3 years. Participants were recruited from parents attending a virtual meeting; 66 of 225 (29.3%) parents signed up to participate in the interviews. Members of this subsample were contacted and scheduled to be interviewed until theoretical saturation was reached [39]. Thirty-eight participants lived in one of 34 census tracts in low-income inner-city neighborhoods of Los Angeles, California; one participant lived in a low-income census tract in the northwestern part of the city, and one participant lived in a census tract in a suburban community east of Los Angeles. A description of the program and procedures for participant recruitment can be found in our earlier study [39]. All but one of the participants were female, with an average age of 42 (S.D. = 7.4) years. Sixty-two percent had a high school education or less; the remainder had one or more years of college Two-thirds of the participants were Hispanic/Latinx; 22.5% were Asian American, 2.5% were African American, and 7.5% were non-Hispanic white. More than half (57.5%) were employed outside of the home, and 72.5% rented their place of residence. Participants resided in their current neighborhoods for an average of 15 years. The proportion of Hispanic/Latinx participants was comparable to the percentage of Hispanic/Latinx residents of the 36 census tracts where participants resided. In contrast, Asian Americans were over-represented in the study sample (22.5% vs. 8.3%), and adults with less than a high school education were under-represented (22.5% vs. 37%). Each parent received a USD 30 Visa or Amazon gift card as compensation for their participation.

### 2.3. Data Collection

Quantitative measures of air pollution and socioeconomic vulnerability of the census tracts where study participants resided were obtained from the Office of the California Environmental Protection Agency’s *CalEnviroScreen* [42], which gives census tract-specific percentiles based on state distribution of several pollution and population characteristics indicators. Air pollution indication indicators used in this study included the following: the amount of daily maximum 8-h Ozone concentration, annual mean PM_2.5_ concentrations, diesel PM emissions from on-road and non-road sources, toxicity-weighted concentrations of modeled chemical releases to air from facility emissions and off-site incineration, and traffic density in vehicle-kilometers per hour per road length, within 150 m of the census tract boundary. Population characteristics included the following: age-adjusted rate of emergency department visits for asthma, percent low birth weight, age-adjusted rate of emergency department visits for heart attacks per 10,000, percent of the population over 25 with less than a high school education, percent limited English speaking households, percent of the population living below two times the federal poverty level, percent of the population over the age of 16 that is unemployed and eligible for the labor force, and percent housing-burdened low-income households. We also examined the average percentile of three summary cores: (1) an average of percentiles from the Pollution Burden indicators (with a half weighting for the Environmental Effects indicators), scaled with a range of 0–10; (2) an average of percentiles from the Population Characteristics indicators, also scaled with a range of 0–10; and (3) *CalEnviroScreen* Score, which is the Pollution Score multiplied by Population Characteristics Score [42]. Census tract data on the proportion of residents who were Hispanic/Latina/Latinx, Asian American, and with less than a high school education available were used to assess how representative study participants were of their neighbors.

Using a semi-structured interview guide based on our conceptual framework [39], all interviews were conducted in English, Spanish, and Mandarin by trained bilingual research assistants using the Zoom platform or by telephone. Participants provided information on the following: (1) the threat of wildfire smoke and other forms of air pollution, including past experience with wildfire smoke and days of poor air quality, perception of current levels of air pollution in the community compared to 10 years ago, reasons for a perceived increase in air pollution, perceived health impacts of air pollution and whether some people are more vulnerable to such impacts than others, and concerns about the health of your children during a day of poor air quality; (2) adaption strategies and resources used or recommended for adapting to air pollution, including advanced notifications and preparations, changes in daily activities, and experience, if any, seeking medical care during a day of poor air quality; and (3) impact of air pollution on the physical and mental health of participant or family members.

Participants were interviewed for 45–60 min between late November 2021 and early January 2022. To ensure their accuracy and validity, interviewers provided summaries of responses to questions from participants throughout the interview. The Institutional Review Board of the University of Southern California reviewed and approved all study activities.

### 2.4. Data Analysis

Quantitative data from the *CalEnviroScreen* for each of the 36 census tracts where study participants resided were used to create three measures of exposure and demographic vulnerability of the study participants to air pollution for each pollution and demographic characteristic indicator: mean percentile, lowest percentile, and highest percentile based on the state distribution of each indicator.

Digital recordings of semi-structured interviews were analyzed using a thematic content analysis method [43] involving seven steps: (1) transcription of interviews with English-speaking participants using Zoom AI software and interviews with Spanish- or Mandarin-speaking participants using the Sonix AI web-based services; (2) reading and familiarization of transcripts and interviewer notes to check the accuracy and a achieve a broad understanding of content; (3) open and axial coding of text segments [44] using the NVivo20 computer program; (4) construction of themes through comparison and contrast [45] (5) reviewing themes, (6) defining and naming themes; and (7) finalizing the analysis. Each text was independently coded by at least two investigators. Disagreements in the assignment or description of codes were resolved through discussion between investigators and enhanced definitions of codes. Interrater reliability in coding was assessed by means of a kappa statistic [46]. A more detailed description of the analysis process can be found in our previous study [39].

## 3. Results

### 3.1. Pollution Exposure (Risk)

Census tract data available from the California Office of Health Hazard Assessment indicates the pollution burden and socioeconomic vulnerability of participant neighborhoods are among the highest in the state (within the top 15 percentile of all census tracts within the state, with some indicators such as PM_2.5_, the airborne release of toxic chemicals, percent with less than a high school education, percent living two times below the poverty level, and percent living in housing-burdened low-income households in the top 20% (See Table 1). Some residents lived in census tracts with indicator percentiles that are among the highest 1–3% of all census tracts within the state. All but six participants (85%) lived in census tracts that were above the 75th *CalEnvironScreen* score percentile that met the designation of “disadvantaged community” and were eligible for state adaptation funds per California Senate Bill 535 [47].

### 3.2. Threat Assessment of Air Pollution

All participants reported having been exposed to wildfire smoke and other forms of air pollution in the past year when the air quality was especially bad, ranging from a few days to a month, and 38 participants (95%) asserted that there was more air pollution now than there was 10 years ago (Table 2). The three most commonly cited reasons for the increase in air pollution were vehicular traffic (*n* = 15), lack of concern over the environment (*n* = 12), and production and release of chemical and industrial contaminants (*n* = 11). As described by one participant, “But I think one of the reasons is that there are many, many factories, there are many cars. People throw garbage everywhere. So that’s why all of this has affected the air.”. Other factors responsible that were cited by participants included wildfires (*n* = 10), overpopulation (*n* = 5), deforestation (*n* = 4), and rising temperature and drought due to climate change (*n* = 4). All participants were able to identify one or more health risks associated with air pollution, including asthma and other respiratory diseases, cardiovascular disease, allergies, depression, and anxiety. Young children, older adults, people working outdoors, and persons with pre-existing chronic conditions were perceived to be at the greatest risk for these health impacts. Twenty-five participants expressed concerns about the effects of air pollution on the health of their children; half of these participants (*n* = 12) had children with asthma or severe allergies. Participants were particularly concerned about the potential health impacts of wildfire smoke on the health and well-being of their children. This was illustrated by a comment provided by one of the participants:

“There was a fire. And then you could smell it. And then all the smoke, you could smell it. And all the ashes were falling on the car. I can’t remember when. It was during the summertime and the car was so dirty. And I kept on having to go through the car wash. And I was concerned with when we were getting in the car and the kids, they wanted to touch everywhere. Oh, it’s ashy. Let me write my name on the car or let me just touch it. And I’m like, ‘No, stop, it’s dirty.’ And then as soon as we get in the car, I have to clean their hands off. And I was concerned about that”.

However, four parents stated they were not worried about the effects of air pollution on the health of their children because they kept them indoors on days of poor air quality. As one parent commented, “The knowledge that right now there is no good [air] quality only helps me to be more sensitive that I should be more careful with the children. No, it is better to take more precautions and not go out or do activities outside until the pollutants go down a little bit.”.

### 3.3. Air Pollution Adaptation Behaviors (Response)

The most common behavior reported during a day of poor air quality was staying indoors. This was reported by 60% of the participants (Table 3). This was followed by wearing facemasks (37.5%) when outdoors, keeping doors and windows closed when indoors (25%), and checking the daily air quality index (17.5%). Six participants also reported using fans and portable AC units to circulate the air indoors, and 22 participants had reported using them to keep cool on days when poor air quality coincided with warm temperatures. Seven participants reported the practice of drinking plenty of fluids during days of poor air quality to reduce inflammation in the throat. One of the Chinese participants mentioned the consumption of teas or traditional soups to cleanse the body of toxic chemicals: “Wildfire weather in the summer, right? We Guangzhou people, just drink more herbal tea, soup, and go out less”. Another Chinese participant explained that many of the traditional methods for adapting to heat in China could also help with adapting to air pollutants: “The mung bean soup is to clear away heat and detoxify. We think people have some heat in their bodies. If you want to set fire to it, what about bitter gourd and winter gourd? It’s the kind of heat… Uh, chrysanthemum, and the honeysuckle to drink with water? It can also clear away heat and detoxify”. Smaller numbers of participants recommended avoiding physical activity outdoors (*n* = 3), planting trees on their property (*n* = 2), changing their clothes after being outdoors for prolonged periods (*n* = 2), and purifying indoor air with boiled eucalyptus leaves. Fourteen participants (35%) reported they would seek medical attention if necessary.

### 3.4. Air Pollution Adaptation Resources

As illustrated in Table 4, study participants had few resources to support their adaptation efforts. The most widely available resource used by participants when going outdoors was the face mask used to protect themselves from coronavirus infection. All participants possessed one or more face masks because of the COVID-19 pandemic. However, as noted earlier, only 15 participants reported wearing them specifically on days of poor air quality, and face masks were perceived by some participants as having limited utility to wildfire smoke:

“And during that time, we were already wearing the mask. So, I was happy that we were all wearing the mask. But you could still smell it and just the fact that it was everywhere, you can see that it was falling on you. But I know it was falling. It was still the ashes were still falling because it was right there on the car. So, I was concerned about that when there were the fires going on”.

Another resource widely available to participants was fans and air conditioning units to circulate indoor air. Most participants (77.5%) reported possession of fans, while 62.5 percent of households also possessed air conditioning units, which could be used to circulate and filter the air as well. The use of air filters was cited by only two participants.

### 3.5. Health Effects of Air Pollution (Results)

Participants reported several different types of physical and mental health impacts experienced during days of poor air quality (Table 5). Thirty-three participants (82.5%) reported symptoms of physical and/or mental health problems during a period of poor air quality, most of which (92.5%) were due to wildfire smoke. Twenty-three participants (57.5%) reported some health problem or physical discomfort during days of poor air quality, including sore throat or swollen eyes (*n* = 14), allergies (*n* = 6), dermatological conditions (*n* = 2), and asthma or trouble breathing (*n* = 4). Fifty percent of participants also reported feeling depressed or anxious during days of poor air quality, much of which was related to concerns about the health of their children, their own health and feelings of discomfort, and confinement indoors. Participants also reported health problems experienced by their children, including asthma (*n* = 9), allergies (*n* = 4) or sore throat, and swollen eyes (*n* = 6). For the most part, the physical symptoms were minor and, if necessary, were treated with over-the-counter analgesics. Only three participants reported seeking medical attention for difficulty breathing or severe allergic reactions to wildfire smoke. Similarly, symptoms of depression or anxiety were short-term and mild.

## 4. Discussion

Every participant in our study reported exposure to wildfire smoke and other forms of air pollution in the previous year, lasting from a few days to a few months. All but two of our participants asserted that this exposure had increased in frequency and intensity over the past decade as a result of increases in vehicular traffic, population, chemical emissions from industrial sources and other forms of human activity, wildfires and deforestation, and a lack of concern for the environment. Apart from the increase in wildfire smoke in recent years [22], these findings are not consistent with studies documenting a decline in PM_2.5_ in the Los Angeles Basin over the past 40 years [48]. However, these individuals have a higher likelihood of living in areas of the US with greater exposure to air pollution, and its impacts occur due to discrimination and segregation [22,49,50]. As noted earlier, 85% of the participants in this study lived in census tracts that were refined as vulnerable communities based on a *CalEnvironScreen* score above the 75th percentile of census tracts in the state of California. The majority of study participants (72.5%) rented their place of residence, which placed them at increased risk for indoor air pollution (22–24).

Our findings also revealed a common concern about the likely impacts of days of poor air quality on the physical and mental health of participants and their children. Especially those with a history of asthma who are especially vulnerable to air pollution [51,52]. However, five parents expressed confidence in their ability to protect their children from exposure to wildfire smoke and other forms of air pollution.

Adaptive behaviors to days of poor air quality reported by participants fell into two groups, indoors and outdoors. The first group of behaviors included remaining indoors to avoid wildfire smoke and other forms of air pollution, keeping doors and windows closed, using fans or air conditioning units on the air setting to circulate indoor air, and using air filters. Staying indoors and limiting outdoor physical activity have been recommended for extreme air pollution events [53,54] and are widely practiced in the United States [53] and elsewhere [55,56]. However, several studies have demonstrated the limited effectiveness of this strategy, as community exposure to wildfire-associated PM_2.5_ can occur in both outdoor and indoor environments [20,21]. Outdoor behaviors included wearing face masks, planting trees, and changing clothes when returning from being outdoors. Behaviors performed both indoors and outdoors included consumption of liquids and daily monitoring of the air quality index.

These adaptive behaviors were facilitated by the availability of fans and air conditioners to circulate indoor air and facemasks purchased during the COVID-19 pandemic that could also be used to mitigate exposure to smoke/air pollution when outdoors. Although air filters have been widely recommended as a resource for days of poor air quality [54,57], only two participants mentioned having purchased them, and only one participant mentioned the purchase of an air purifier. In contrast, 60% of study participants reported receiving advanced warnings of days of poor air quality due to wildfire smoke and other forms of air pollution from news reports and smartphone apps that report daily air quality index levels and warnings to residents at risk for respiratory illnesses.

Several recommendations have been offered for reducing health disparities related to exposure to air pollution in general and wildfire smoke in particular. These include public investment in developing greenspace in urban settings [58], free/low-cost air filters and high-quality N-95 masks for low-income households, a clean air community space, and informational and educational campaigns during wildfire smoke events [36,51,53,54,55]. The latter is especially noteworthy as it has become a consistent refrain in the literature [52,53,59]. Government agencies have responded to this call by making information available to the general public. For instance, the Environmental Protection Agency provides recommendations on how to reduce exposure to indoor air pollution [60]. However, the results of this study suggest that residents of low-income urban neighborhoods are well aware of the threat posed by wildfires and human action to health and well-being, especially among children, older adults, and people with pre-existing chronic health problems. This awareness can be attributed to prior experience with symptoms of physical and/or mental health problems during days of poor air quality and exposure to public health warnings about the potential hazards associated with these events. Nevertheless, these findings suggest a call for policies designed to improve communication about which resources provide the greatest protection from hazardous air quality due to wildfire smoke and other forms of indoor and outdoor air pollution. For instance, residents in vulnerable communities would benefit from knowing that HEPA air filters do a much better job than fans or non-HEPA air filters or that cloth face masks are not nearly as effective as N-95 masks.

Despite the exercise of adaptive behaviors and availability of resources to mitigate exposure to air pollution, 25 study participants (62.5%) reported one or more symptoms of physical discomfort experienced by themselves or their children during a period of poor air quality. An association between wildfire smoke in particular and air pollution in general and increased rates of respiratory and cardiovascular disease has been extensively documented in previous research [27,28,29]. However, although the majority indicated a willingness to seek medical attention if necessary, only three of the participants sought treatment for these problems. In addition to the possible effects of their adaptation behaviors and resources, this might be attributed to the fact that the symptoms experienced were relatively mild and short-term and that few participants possessed any predisposing health conditions that might be exacerbated upon exposure to wildfire smoke or other forms of air pollution. Limited use of healthcare services during periods of poor air quality may have also been due to limited access to such services. Some participants also reported that the exercise of adaptive behaviors minimized concerns about the health and safety of their children.

The high levels of anxiety and depressive symptoms reported by study participants are consistent with the findings from other studies of stress experienced in communities exposed to wildfires or wildfire smoke [31,32,36,56,59,61]. However, as these symptoms were not severe enough to warrant clinical intervention, a stepped-care approach to the delivery of mental health services by nonprofessionals is recommended [62,63]. This approach has been found to be effective in addressing symptoms related to wildfires [64] and other extreme weather events [65]. Moreover, these symptoms were reported in inner-city neighborhoods that have not been directly threatened or forced to evacuate due to the proximity of wildfires. Our study participants were more concerned about confinement indoors and the smoke-related threat to their own physical health or that of their children than they were about property damage, forced evacuation, and disrupted livelihood and recreational activities. Other studies have demonstrated an association between depression and airborne particulate matter not related to wildfires [66,67,68]. This suggests a focus on policies that target local sources of non-wildfire-related air pollution (regulation of industrial and vehicular emissions, expansion of accessible greenspace) and prevention of physical health impacts associated with all forms of air pollution [49,69].

The public health significance of many of these findings must take into consideration limitations in our study design. Study participants were drawn from a small, nonrandom sample of residents of low-income neighborhoods in Los Angeles, California; participants also had higher levels of education than the average for their neighbors. Thus, the findings may not be generalizable to all low-income urban dwellers, including those living in the same census tracts. Nevertheless, participants who lived in census tracts identified as having high pollution burden and socioeconomic vulnerability and thus representative of residents of other low-income neighborhoods with similar environmental and sociodemographic characteristics. It may also suggest that the risk and responses to wildfire smoke and other forms of air pollution in these census tracts are not uniformly distributed among their residents.

Second, women comprised almost all of our study participants. Given that women are more likely than men to experience physical and mental health problems related to wildfire smoke [70,71] and to engage in adaptive behaviors when exposed to extreme air pollution events [55], future research should also include sufficient samples of men.

Third, data collection was not intentionally timed to coincide with a specific acute event of poor air quality (e.g., during a wildfire or a heat wave). Consequently, many of the reports of past experiences with days of poor air quality were retrospective and subject to recall bias. However, with few exceptions, reports of exposure to wildfire smoke and adaptation behaviors were based on experiences in the previous six months of participants’ interviews.

Fourth, any presumed associations between health status and exposure to wildfire smoke and other forms of air pollution are constrained by the absence of objective measures of either set of variables. Longitudinal assessments of physical and mental health status and levels of exposure to specific components of polluted air among residents in low-income urban dwellers using objective and validated instruments are highly recommended.

Finally, this study was designed to identify potential hypotheses but not to test them. Although our qualitative findings point to the existence of associations between levels of socioeconomic and environmental vulnerability, adaptation behaviors, resources, and health outcomes, they must be verified with larger samples in quantitative studies.

## 5. Conclusions

Low-income residents of urban settings perceive themselves to be at risk for the physical and mental health impacts of wildfire smoke and other forms of air pollution. They have adapted to that risk by monitoring air quality, staying indoors, using fans and air conditioning to circulate indoor air, or wearing face masks when outdoors. Such adaptive behaviors may have contributed to low levels of health services utilization despite widespread reports of physical and mental health symptoms. In addition to policies and programs designed to reduce air pollution at its source, policies that facilitate ongoing surveillance of air quality and promote the development of community-level resources and responses, such as the expansion of greenspace and limited outdoor activities at schools, as well as individual-level resources such as air filters and respirators, would appear to be most responsive to current needs for the prevention and mitigation of adverse effects to health and well-being in these vulnerable communities.

## Figures and Tables

**Table 1 ijerph-20-05393-t001:** Air pollution burden and demographic characteristics of 40 study participants residing in 36 census tracts in Los Angeles, CA.

	*CalEnvironScreen* Score Percentile		
Indicator	Mean Percentile	Minimum	Maximum
Ozone	50.59	35.16	97.00
PM2.5	80.86	65.95	93.25
Diesel	59.60	24.49	99.42
Toxic release	80.67	58.88	91.86
Traffic	64.80	32.70	99.48
Total pollution burden score	73.82	40.66	96.76
Asthma rate	73.86	7.60	97.91
Percent low birthweight	73.10	0.98	99.28
Heart attack rate	64.93	5.50	97.43
Percent less than high school education	85.12	45.58	98.17
Percent limited English-speaking households	76.98	30.69	98.32
Percent of population living below two times the federal poverty level	80.40	31.55	99.80
Unemployment rate	66.79	8.69	97.42
Percent housing-burdened low-income households	83.56	29.72	99.58
Total demographic characteristics score	84.22	29.50	99.48
CalEnviroScreen Score	85.21	45.84	97.06

**Table 2 ijerph-20-05393-t002:** Air pollution threat assessment.

Code	N	Illustrative Quotes
Recent exposure to wildfire smoke and/or other forms of air pollution	40	“In this way, look at the mountain fire, even if it is very far away. Even if we live in LA, it is very hot in Chino. Even if it is our side, the air quality is bad. It will get bad. At least a week, at least”. “But this past summer. It was about two weeks where we didn’t see the sun and we were always told that the air quality was very bad… There were even three days where they said that we had to wear the masks all the way in so as not to breathe, especially those of us who suffer from respiratory diseases”.
Trend in past 10 years	38	“I totally think there’s more. We have, I never paid attention to it other than when you see it in the air until we went to China. In China, you can see it. You can feel it in the air, almost. It’s like this nasty stuff like that you can touch. And when we came back, we would talk when we were over there and be like, I can’t wait to go back to California, where I can breathe the fresh air. And it was really strange because even though we’re in a pandemic and people are using their cars less. Of course, there were a lot of days where we had high AQI, and you could see it and I know you know the fires and different things that happened, but we would talk about it, me and my husband like “oh my gosh maybe we didn’t know what we were talking about” like, there’s all sorts of days where the API is really high and we’re getting these alerts. And so, it’s definitely better than China but I think we’re, I think because of the ships out in the Marine in the port, all the backups and everything I really think that our air pollution this year has been really bad. And so, I do think it’s worse.
Reasons for increase		
Vehicular traffic	15	“I feel like the cars, all the old cars that are just, you know, not up to date getting all these, like, illegal smog checks, you know. And so, you know, I see it every day by our neighborhood. All these cars are just like throwing out all this smoke…” “There’s also a lot of pollution, through a lot of family cars. Pollution, especially here in LA. There’s a lot of traffic; there’s a lot of cars. Well, yeah, that adds to the pollution”.
Lack of concern for environment	12	“Everything plastic, everything that one leaves, everything heats up and everything goes up into the atmosphere. And that’s where I think that causes harm because we ourselves cause it. And like the oil, car oil, the same it evaporates, it drains. This is my understanding of it”. “I don’t know what kind of people do that, but it became very fashionable that they have set fire to garbage cans and that plastic that the garbage throws such an ugly black smoke too”.
Release of industrial and chemical contaminants	11	“There are many factories that make pure smoke, smoke and smoke”.
Wildfires	10	“But if you say that the air quality is not good, it is basically that the days of the wildfire it will be very poor”.
Overpopulation	5	“And then today we are also overpopulated, and I think we generate too much waste and that does not help to improve pollution”.
Deforestation and lack of greenspace	4	“Well, I don’t see that many trees. I don’t see so many trees anymore. Well, not anymore, not as recently as now… And I think that’s because they are building more buildings there are not so many anymore”.
Climate change	4	“Oh well, that’s because they say it doesn’t want to rain here in California, and you see it hasn’t rained much right now. I think all that affects the air and there have been many fires” “The increase in temperature has led to an increase in wildfire and air pollution”.
Consequences of increase		
Anticipated health problems	33	“It does have a strong relationship, especially with respiratory problems with diabetes, obesity, anxiety. It is closely related”. “Yes, more those who are directly exposed. In our neighborhood, we are exposed to the oil wells or the freeway, which is a highly polluted area. All these agents are creating a deterioration in our health physically and also mentally because we have consequences or side effects. There is an effect on our mind”. “Well, I think asthma is one for him. For me it is one of the main ones that causes asthma the most, environmental contamination. Because it is everything that is absorbed by the body and goes directly to the lungs. Apart from that, it could if people have heart problems”. “I know air pollution in general like it’s bad for your lungs. And I know that it can affect your allergies, depending on what’s in the air. I know, I know that it can get into other things like if the air is polluted than it can get into our water. And personally, I know you didn’t ask but when I see bad air pollution it depresses me”.
Concerns about children	25	“I’m afraid it will affect his growth. If breathing pollution enters one’s body, then of course it is not good for growth”. “I’m worried about if they breathe in something very bad at this time in the future, they’re going to create more health problems for them when they get older.

**Table 3 ijerph-20-05393-t003:** Air pollution adaptation behaviors.

Strategy	N	Illustrative Quotes
Stay indoors	24	“Yeah, we usually watch TV or movies, but it was hard because summers have their long days. So especially when we have kids, they want to go outside. I have a four year old and she wanted to go outside. We couldn’t smell like smoke”. “Well, we stayed at home and played as a family because we could not do anything outside because there was a fire”.
Use fans for air circulation	22	“Yeah, we try to keep the doors keep the doors closed, the windows closed, and just have the fan on”.
Wear face masks	15	“Yes, don’t go outside, use masks for the nose and mouth to avoid breathing bad air”. “Sometimes we feel like we should just wear the masks because the air quality is just bad”.
Close windows	10	“Well, the only thing we did was to close the windows and doors so that the smoke could not enter or the smell of smoke”.
Check air quality	7	“I’m mostly like I mentioned before, mostly because of my son we had to be cautious about, you know, places to go out and stuff like that or constantly checking the weather. I mean the air quality and the areas and stuff like that”. “And checking air quality. And I guess it would be just having the space available for the kids to be able to still kind of enjoy their school day without having to be outside and just having activities for them. Yes, I know. You know, when they have to stay indoors, it’s not fun for them. So, you know, just making it a fun time”.
Consume liquids	7	“During the time of the wildfire, the throat was sometimes very dry. Interviewer: Dry throat. Uh, what do you do to ease the throat? CPQID 5: Drink more water. Uh, even if you go to the doctor, drink more water, go back and drink more water, that’s all”. “We just kept inside the house and only drank a lot of liquid”.
Use air conditioning for air circulating and filtering	6	“And so, it was a time when there was a fire nearby. It had to be 21 days during, maybe sometime, during the fall. Yeah, it was windy and then, the air quality was so bad that you can see the sun was like all red. Yeah, there were ashes everywhere. And I remember we had to keep all the windows closed, and actually turn on just a fan on the central AC, not to make it cold, but just a fan to circulate”.
Avoid physical activities outdoors	3	“Well, the only thing that we did was not be very exposed to the open air and we didn’t do outdoor stuff, like running, playing, or something like that”.
Plant trees	2	“We have been trying to plant trees around our house. Because I know that the more trees we have, the cleaner the environment will be”.
Change clothes	2	“Then breathe on that, as if you go out and feel dirty as soon as you come back. You have to change clothes soon. You can’t keep your clothes indoors or outdoors for too long. If you come back, you have to change your clothes and wash them immediately. Then it feels dirty.
Purify air with eucalyptus	1	“Well, I think for the poor quality it was necessary to keep us inside the house, try to at least try to keep away from breathing out there. In our case it is what we sometimes do for supposedly to clean the air. We boil what is eucalyptus to purify the environment. In a pot we put water and add eucalyptus branches”.
Seek medical attention if necessary	14	“I would go directly to my doctor at the clinic, and he would have to make a diagnosis. And from that diagnosis, see if I have to be referred to a center, depending on the one that I find myself in. If it is a chronic respiratory disease, we will have to seek help to see a specific specialist but depending on the diagnosis”.

**Table 4 ijerph-20-05393-t004:** Air pollution adaptation resources.

Resource	N	Illustrative Quotes
Masks	40	“I normally wear a mask when I go outdoors on days like that since, you know, we’re already wearing masks because of the pandemic. I did purchase the N95 masks for my son, so that he was in less contact with the air on days when the air is bad so that it didn’t affect him. But throughout that process like he was perfect, nothing happened to him. [I’m] not sure if it was the mask or just, you know, the places that we choose to go and stuff like that”.
Fans	31	“I have a fan because the air conditioning, I feel like, doesn’t re-circulate all the air”.
Advanced warning/notifications	24	“Well, when they warn in the news to be aware of the air, when they warn that there is bad quality or there is a lot of pollution that day because they always warn that today is going to be a time, it is going to be tomorrow, the air is going to be polluted”. “Well, the fire lasted… It was a good one month because I remember I had to check on the air quality every day, some type of app that tells you by the air quality whether it is safe to go out. I remember checking on it for four months. And then finally it ended”.
Air conditioners	25	“You know how the central AC has like a fan, you can just turn on the fan. Yeah. Okay, so the condenser isn’t working. Just a fan blowing, and then it helps to filter the air”
Social networks	5	“I heard from friends that if the air is not good, it’s easy to get asthma”.
Air filters	2	“Put an air filter at home. I just bought it last year, I forgot when, I bought it on Amazon”.

**Table 5 ijerph-20-05393-t005:** Health impacts of air pollution.

Health impact	N	Illustrative Quotes
Physical health	25	
Eye/throat irritation	14	“Yes, because then we felt it in our throats, it burned all over”. “It smelled. The … my eyes were irritated a little bit, for the first day or so. So that’s when I really tried to avoid going outside”.
Allergies	5	“My allergies? Yes. Regularly, when that happens, I suffer a lot of… a lot of coughing. My eyes water, my nose gets inflamed and it’s like this. The more the air is polluted, the more I suffer from it”.
Difficulty breathing	5	“Yes. Yes. Because it starts with allergies, and then I start coughing. And then I see that I’m struggling for air, so I had to use my inhaler”.
Rashes	2	“There was, or climate and all of that yes like itchy pimples. How do you say? Allergy. Yes, that. Like an allergy, like dryness. They were kind of a big deal, but dryness”.
Anxiety or depression	20	
Due to concern for children	7	“It made me anxious. Yeah. It made me anxious because I have a daughter that has asthma”.
Due to potential health impacts	4	“And apart from the physical symptoms, it made me feel more depressed or anxious, like it’s scary to go out because you’ll get sick, so things like that”. “My concern was about our health in general”.
Due to physical discomfort	3	“A little bit more depressed because that’s how the discomfort felt in the body”.
Due to wildfires	3	“I mean, of course, like thinking of like what the sky looked like at that time. Seeing all the ash and really feeling like, you know, the impact of those fires, I remember feeling anxious like that”.
Due to uncertainty	3	“Yes, anxious because of the uncertainty that it would continue like this”.
Due to confinement	3	“It did really just make me feel more anxious, because I think around that time, like our kids play soccer so we wouldn’t let them go to practice or go to games or anything. So, it’s kind of just like being cooped up”.
Due to visibility of pollution	3	“I know that it can get into other things like if the air is polluted, then it can get into our water and personally, I know you didn’t ask but when I see bad air pollution, it depresses me”.

## Data Availability

The data presented in this study are available on request from the corresponding author.

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
