# Peer review of "Adaptation Resources and Responses to Wildfire Smoke and Other Forms of Air Pollution in Low-Income Urban Settings: A Mixed-Methods Study"

_ijerph, 2023, doi:10.3390/ijerph20075393_

Round 1

Reviewer 1 Report

It is not easy to connect the environmental issue with the real reactions of the people, especially for those with low income who have less opportunity to understand the impact of air pollution and the access to protection of themselves. So it is very important for air pollution researchers and decision makers with comprehensive thinking to decrease the adverse impact of air pollution to those people.

The ways to get the data, to analysis and the discus, and to get the conclusion are reasonable. For the poor people, this is what this paper moved me a lot too.  The work is a very good to supplement to other researches in terms of all kind of people, not only well beings but those in poor conditions.

One minor question in the introduction:

High teperature maybe one of the reason fro wild fire but ii is not the casue of air pollution. Please correct it. 

Author Response

We wish to thank the reviewer for the positive assessment of the manuscript. 

High temperature maybe one of the reason fro wild fire but ii is not the cause of air pollution. Please correct it.

The reviewer is correct that high temperature is not a cause of air pollution, but it does contribute to higher concentrations of air pollutants in certain areas. WE apologize if our statement implied such a causal association. To clarify the association, we had edited the following:

In turn, although not a cause of air pollution, increases in ambient air temperatures have led to greater concentrations of airborne pollutants such as ozone (O3) [3-6] and fine particulate matter (PM2.5) [7,8] in many parts of the world, especially in wildfire prone areas [9]; however, the associations between temperature and levels of O3 and PM2.5  exhibit regional variability [10]. 

Reviewer 2 Report

This paper shows the results of a qualitative study focus on how the low-income inhabitants of a big city react in situations of high air pollution.

The study is interesting since, as the authors mention, there are only few studies about the knowledge, attitudes, behaviours and resources to mitigate the impacts due to the air pollution in communities with low incomes.

My general opinion is that the paper is well written and explained. However, I have some concerns:

Regarding the introduction, the authors just mention the outdoors air pollution without considering the problem of the indoor air pollution mostly in situations of pollution warnings. There are a lot of webinars, leaflets, notes etc in the EPA website that are not mentioned and those are free resources that can be suggested to the people for using. Therefore, some additional information about indoor air pollution in the introduction should be added.

When the authors describe the participants, there is no mention to gender, age or study level. It is true that they mention in the Results that all but one of the participants were women. Nevertheless, it is important to give those information in the Material and Methods not only in the Results. Even though the authors have another paper with a deep description, a brief one about the program and procedures should be helpful to fully understand the study.

Finally, the Discussion is well structured and explained. Again, I recommend some lines mentioning indoor air pollution. For instance, the authors mention in line312 the use of HEPA filters.

In line 316, authors mention that 62.5% of the participants reported symptoms of health problems when during all the paper they are indicating the number of participants that are showing some behaviour or attitude. Please, also add the number of participants that is that 62.5%.

In conclusion, the paper is ready to be published with a minor revision.

Author Response

This paper shows the results of a qualitative study focus on how the low-income inhabitants of a big city react in situations of high air pollution.

The study is interesting since, as the authors mention, there are only few studies about the knowledge, attitudes, behaviours and resources to mitigate the impacts due to the air pollution in communities with low incomes.

Response: We thank the reviewer for this positive assessment of our manuscript.

My general opinion is that the paper is well written and explained. However, I have some concerns:

Regarding the introduction, the authors just mention the outdoors air pollution without considering the problem of the indoor air pollution mostly in situations of pollution warnings. There are a lot of webinars, leaflets, notes etc in the EPA website that are not mentioned and those are free resources that can be suggested to the people for using. Therefore, some additional information about indoor air pollution in the introduction should be added.

Response: We have added the following in the introduction:

These residents are also vulnerable to indoor pollution linked to wildfires [20-22] and other forms of outdoor pollution due to increased poverty and substandard housing [23, 24]. Although the indoor environment is often overlooked in relation to environmental health in general [25] and climate-related health in particular, studies have found the between 80 and 90 percent of time is spent indoors [26], and this is likely to increase during wildfires due to public health warnings about the risk of exposure to smoke [22].  

We now also mention the resources available in the EPA website in the discussion section.

When the authors describe the participants, there is no mention to gender, age or study level. It is true that they mention in the Results that all but one of the participants were women. Nevertheless, it is important to give those information in the Material and Methods not only in the Results. Even though the authors have another paper with a deep description, a brief one about the program and procedures should be helpful to fully understand the study.

Response: We have placed the following information in the Methods section under the participants section:

All but one of the participants were female, with an average age of 42 (S.D. = 7.4) years. Sixty-two percent had a high school education or less; the remainder had one or more years of college Two thirds of the participants were Hispanic/Latinx; 22.5% were Asian-American, 2.5% were African American, and 7.5% were non-Hispanic white.  More than half (57.5%) were employed outside of the home, and 72.5% rented their place of residence. Participants resided in their current neighborhoods an average of 15 years. The proportion of Hispanic/Latinx participants was comparable to the percentage of Hispanic/Latinx residents of the 36 census tracts where participants resided. In contrast, Asian-Americans were over-represented in the study sample (22.5% vs 8.3%) and adults with less than a high school education were under-represented (22.5% vs 37%). 

We now summarize the procedures for sample selection and recruitment of participants in this section as well.

Finally, the Discussion is well structured and explained. Again, I recommend some lines mentioning indoor air pollution. For instance, the authors mention in line312 the use of HEPA filters.

Response: We have added the following:

The majority of study participants (72.5%) rented their place of residence, which placed them at increased risk for indoor air pollution (22-24).

We have also clarified the need for policies that reduce both indoor and outdoor pollution.

In line 316, authors mention that 62.5% of the participants reported symptoms of health problems when during all the paper they are indicating the number of participants that are showing some behaviour or attitude. Please, also add the number of participants that is that 62.5%.

Response: We have added the number as requested.

In conclusion, the paper is ready to be published with a minor revision.

Reviewer 3 Report

Air pollution including wildfire smoke affected human health both physically and mentally. Otherwise, how low-income residents in urban area affected by these pollutions keeps unknown. This study aims to investigate the adaptation resources and responses to wildfire smoke and other forms of air pollution in low-income urban settings based on a mixed-methods study. The result will provide very meaningful support for both relative policies and programs designing to reduce air pollution and individual-level resources for prevention and mitigation of adverse effects to health and well-being in vulnerable communities.

Only a small suggestion provided here:

In Materials and Methods, for “2.1 Design” part, authors referred to “We operated from a conceptual framework that places multi-step causal chains associated with climate change within a context of socioeconomic and demographic factors, societal actions, and other non-climate drivers.” Could you please clarify the steps in detail?

I think the paper has been well written and could be published after a minor revision.

Author Response

Comment: Air pollution including wildfire smoke affected human health both physically and mentally. Otherwise, how low-income residents in urban area affected by these pollutions keeps unknown. This study aims to investigate the adaptation resources and responses to wildfire smoke and other forms of air pollution in low-income urban settings based on a mixed-methods study. The result will provide very meaningful support for both relative policies and programs designing to reduce air pollution and individual-level resources for prevention and mitigation of adverse effects to health and well-being in vulnerable communities.

Response: We thank the reviewer for this positive assessment of our manuscript.

Comment: Only a small suggestion provided here: In Materials and Methods, for “2.1 Design” part, authors referred to “We operated from a conceptual framework that places multi-step causal chains associated with climate change within a context of socioeconomic and demographic factors, societal actions, and other non-climate drivers.” Could you please clarify the steps in detail?

Response:  We have added the following to clarify the conceptual framework:

Drawing from the Integrated Climate Change and Health Indicator Systems Framework [41], the 4R framework has four components: risk of exposure to hazardous environmental conditions, in this case, days of poor air quality; responses or adaptive behaviors; resources to support adaptive capacity; and results of the health impacts of days of poor air quality. Both responses and resources may be framed within a socioecologic model that includes individual, family, organization and community responses and individual physiological, home and immediate social environment, neighborhood, and mesoscale climate resources.

We also provide in parentheses the 4R categories (risk, response, resources, and results) in the headings of the subsections of the Results.